# Application of Activated Carbon Obtained from Spent Coffee Ground Wastes to Effective Terbium Recovery from Liquid Solutions

**Lorena Alcaraz** [1,]*[**, Dayana Nathaly Saquinga** [2]**, Francisco J. Alguacil** [1][**, Esther Escudero** [1] **and Félix A. López** [1][

[1] National Center for Metallurgical Research (CENIM), Spanish National Research Council (CSIC), Avda. Gregorio del Amo 8, 28040 Madrid, Spain; fjalgua@cenim.csic.es (F.J.A.); mebaquero@cenim.csic.es (E.E.); f.lopez@csic.es (F.A.L.)

[2] School of Chemical Sciences & Engineering, Yachay Tech University, Hacienda San José s/n, de San Miguel de Urcuquí 100119, Ecuador; dayana.saquinga@yachaytech.edu.ec

* Correspondence: alcaraz@cenim.csic.es; Tel.: +34-915-538-900

**Abstract:** A process aimed at the recovery of terbium from liquid solutions using activated carbon (AC) derived from spent coffee grounds (SCG) was assessed. AC was obtained using the hydro-alcoholic treatment of SCG, followed by the physical activation of the as-obtained product. The AC exhibited both microporous and mesoporous structures, which were shown by the corresponding nitrogen adsorption–desorption isotherms and scanning electron microscopy (SEM) images. In addition, a certain graphitic character was found in the micro-Raman measurements. By use of this AC, terbium adsorption was investigated, and the influence of solution pH, temperature, and the adsorbent amount on terbium uptake was tested. In addition, adsorption isotherms and kinetic studies were also evaluated. The best fit was found for the type-1 Langmuir isotherm and pseudo-second-order kinetics model. Thermodynamic studies revealed that terbium adsorption is an endothermic and spontaneous process. Terbium desorption by the use of acidic solutions was also investigated. This work demonstrated that it is possible to recover this valuable metal from liquid solution using the present AC.

**Keywords:** rare earth recovery; low-cost adsorbent; spent coffee wastes; terbium elution

## 1. Introduction

Because of their interesting properties and useful applications, rare earths elements (REEs) have growing attracted attention from the scientific and industrial communities, playing a key role in various sectors such as those of green energy, lifestyle, and defense, among others [1]. These elements are vital components of several electrical and electronic-based materials. However, deposits of natural REEs are limited, leading to a great dependence on secondary sources [2]. Dysprosium (Dy), europium (Eu), neodymium (Nd), terbium (Tb), and yttrium (Y) were considered by the European Commission to be the five most critical REEs in a report entitled "Critical raw materials for the European Union" [3].

Furthermore, due to continuously increasing technological developments, nowadays the production of waste electrical and electronic equipment (the so-called e-waste, or WEEE) is noticeable [4]. As a consequence, different strategies to recover metals and/or REEs from several waste sources are currently being evaluated, for example from Liquid Crystal Displays and Plasma Display Panels (LCD/PDP) [1], mobile phones [1], the automotive industry [5], permanent magnets [6], and phosphors [7]. To achieve this recovery, hydrometallurgical operations with regard to leaching [8], chemical precipitation [9], ion exchange [10], emulsion liquid membrane [11] are under continuous development. However, in most cases these techniques are inefficient for REE recovery, are expensive, and require long time periods [12].

As an effective and low-cost hydrometallurgical process, the adsorption method is a common and useful procedure for REE [13] and metal [14,15] recovery from aqueous solutions and under various media. In this sense, because of their high degree of surface reactivity, porosity, and total surface area, activated carbons (ACs) are well known as effective adsorbents. Nevertheless, these types of materials tend to have a high cost when compared with other types of adsorbents, leading to the need for AC obtention from inexpensive raw materials and through low-cost processes.

Several wastes have previously been reported to obtain ACs. Many of these are biomass-derived wastes, such as winemaking waste [16], seeds [17,18], or even wastepaper and waste cotton [19]. Considering that coffee is an important, widely consumed beverage, and due to the carbonaceous nature of coffee waste which leads to a great number of different residues (pulp, husks, coffee beans, and spent coffee grounds), the manufacture of ACs and the re-use of these coffee wastes is an interesting goal both from economic and environmental perspectives.

In the present work, a process for yielding AC from spent coffee grounds (SCG), and the application of the as-derived carbon as an effective adsorbent to terbium recovery from liquid solutions are described. The AC was prepared by a hydro-alcoholic method followed by a subsequent physical activation process. Several parameters such as solution pH, temperature, and adsorbent amount were investigated to evaluate their influence on the terbium adsorption process. Kinetic and thermodynamic studies were assessed. Finally, once the adsorption process was carried out, terbium was desorbed from the terbium-loaded AC by a desorption process under acidic conditions. The results of this investigation allowed an efficient recovery of this REE element.

## 2. Experimental

### 2.1. Derivation of the AC from the Spent Coffee Ground Waste

The activated carbon used as an adsorbent was obtained from spent coffee grounds (SCG) as a starting material. The obtained waste was recovered after coffee beverage preparation from the canteen of the National Center for Metallurgical Research (Superior Council of Scientific Investigations) in Madrid. The coffee grounds used were a mixture of 10 (wt%) torrefacto roasted, and 90 (wt%) natural roasted coffee. Initially, 45 g of the SCG corresponding solids were subjected to hydro-alcoholic extraction in 600 mL of a solution mixture of 50:50 ($v/v$) ethanol:water, using a Berghof BR3000 reactor at 393 K for 30 min at 50 bars. After cooling down to room temperature, the obtained suspension was filtered in a Millipore Holder filter at a pressure of 7 bars, and the final solid was dried. In this step, a favorable carbonaceous precursor to obtain a final activated carbon was produced. Finally, the precursor was turned into activated carbon (AC) by physical activation. Activation was performed in a rotary kiln consisting of an internal quartz cylinder that was heated using an electric furnace regulated by a PID controller. A carrier gas facilitated the removal of the volatile fraction from the reactor into a condensation set which consisted of 3 consecutive empty glass flasks. For this, 20 g of the precursor were added into a quartz reactor at 1073 K for 120 min. A nitrogen carrier with a flow rate of 0.5 mL/min was pumped into the reactor to act as a gas carrier during the heating ramp. Then, when the treatment temperature was achieved, nitrogen was changed to deionized water, which was introduced in the reactor employing a peristaltic pump with a similar flow rate. The procedure to obtain the activated carbon is shown in Figure 1.

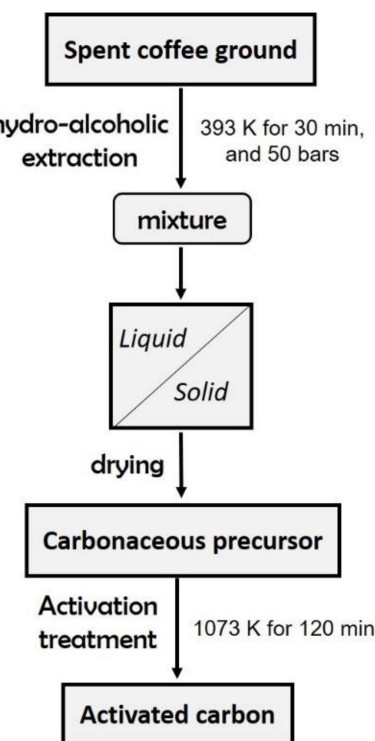

**Figure 1.** Schematic procedure to yield the corresponding activated carbon (AC).

## 2.2. Characterization

The porous structure was characterized by $N_2$ adsorption–desorption at 77 K using an Accelerated Surface Area and Porosimetry System. The specific surface was determined by analyzing the adsorption isotherm via the Brunauer–Emmett–Teller (BET) equation [20] and density functional theory (DFT) models, employing Micromeritics ASAP 2010 and Quantachrome software. The BET and DFT results were compared using the Dubinin equation [21]. The elemental chemical composition and the ash and volatile material contents were determined following the standard methods using a Leco TGA 701 apparatus. The structural characterization was carried out through X-ray diffraction (XRD) using a Siemens D5000 diffractometer equipped with a Cu anode (Cu K$\alpha$ radiation) and an LiF monochromator. Zeta potential measurements were carried out using a Zetasizer Nano ZS apparatus at room temperature. Aqueous suspensions were dispersed with a sonicator with an amplitude of 80% for 300 s in the pH range of 1 to 7. The surface of the obtained AC was examined by field emission scanning electron microscopy (FE-SEM) using the Hitachi S 4800 J apparatus. Micro-Raman measurements were performed at room temperature in a Horiba Jovin Yvon LabRAM HR800 confocal microscope. Raman spectra were recorded under excitation at 632.8 nm line of a He-Ne laser. A charge-coupled device (CCD) detector was used to collect the scattered light dispersed by a 600 lines/mm grating. The spectral resolution of the system used was 1.5 cm$^{-1}$. The terbium (Tb) concentration in the solution was determined by coupled plasma optical emission spectrometry (ICP-OES) using an Agilent 5100 apparatus, with an associated analytical error of $\pm 2\%$.

## 2.3. Adsorption Experiments

Terbium adsorption onto the activated carbon was investigated under batch experiments using a glass reactor with mechanical shaking at 25 °C unless otherwise stated. For the adsorption experiments, the stock solution was prepared by dissolving terbium nitrate hexahydrate ($Tb(NO_3)_3 \cdot 6\,H_2O$) in MilliQ water in order to obtain a 1 g/L solution. Then, 0.025 g of the activated carbon were put into contact with 0.1 L of terbium solution containing 0.005 g/L of the metal (resulting pH value of 5). In order to analyze the influence of the pH, the pH value of the as-described solution was adjusted with 0.1 M HCl until the

desirable pH value was achieved using buffer solutions in a calibrating pH meter. Aqueous samples retired at elapsed times were analyzed and terbium concentration in the solution was determined.

The adsorption capacity ($q_t$, mg/g), and the adsorption percentages (adsorption, %) were calculated using Equations (1) and (2):

$$q_t = \frac{([Tb]_0 - [Tb]) \times V}{m} \tag{1}$$

$$adsorption = \frac{([Tb]_0 - [Tb]_e)}{[Tb]_0} \times 100 \tag{2}$$

where $[Tb]_0$ (mg/L) is the initial concentration, $[Tb]_e$ (mg/L) represents the concentration at equilibrium, and $[Tb]_t$ (mg/L) is the concentration at each time in solution; V (L) and m (g) are the volume of the solution and the mass of the adsorbent used, respectively.

The type-1 Langmuir, type-2 Langmuir, Freundlich, and Temkin linear forms [22,23] (Equations (3)–(6)) are as follows:

$$\frac{[Tb]_e}{q_e} = \frac{1}{q_m \times b} + \frac{1}{q_m} \times [Tb]_e \tag{3}$$

$$\frac{1}{q_e} = \frac{1}{q_m} + \frac{1}{q_m \times b} \times \frac{1}{[Tb]_e} \tag{4}$$

$$\ln q_e = \ln k_F + \frac{1}{n} \times \ln [Tb]_e \tag{5}$$

$$q_e = \frac{R \times T}{b_T} \times \ln A_T + \frac{R \times T}{b_T} \times \ln[Tb]_e \tag{6}$$

where $q_m$ (mg/g) is the maximum adsorbed amount per gram of activated carbon, b (L/mg) is the corresponding Langmuir constant (related to the adsorption energy), $k_F$ (L/g) and n (dimensionless) represent the Freundlich constant related to the adsorption intensity, R (kJ/K·mol) is the universal gas constant, T is the absolute temperature, and $b_T$ (dimensionless) is the Temkin constant.

The kinetic study was carried out by fitting the experimental data to the pseudo-first [24], pseudo-second [25], and Elovich kinetics models using Equations (7)–(9), respectively:

$$\ln(q_e - q_t) = \ln q_e - k_1 \times t \tag{7}$$

$$\frac{t}{q_t} = \frac{1}{q_e^2 \times k_2} + \frac{1}{q_e} \times t \tag{8}$$

$$q_t = \frac{1}{\beta}(\ln \alpha\beta) + \frac{1}{\beta} \ln t \tag{9}$$

where $k_1$ ($min^{-1}$) and $k_2$ (g/mg·min) are the corresponding kinetics constants, and $\alpha$ and $\beta$ are the initial adsorption and desorption rate constant, respectively.

The activation energy ($E_a$) of the process can be calculated from the Arrhenius expression (Equation (10)):

$$k = A \times e^{\frac{-E_a}{R \times T}} \tag{10}$$

where R (kJ/mol·K) is the universal gas constant and T (K) is the absolute temperature.

The standard enthalpy ($\Delta H^0$), the entropy ($\Delta S^0$), and the Gibbs free energy ($\Delta G^0$) were calculated using the following equations:

$$\Delta G^0 = -R \times T \times \ln \left( \frac{[Tb]_t}{[Tb]_e} \right) \tag{11}$$

$$\ln \left( \frac{[Tb]_t}{[Tb]_e} \right) = \frac{\Delta S^0}{R} - \frac{\Delta H^0}{RT} \tag{12}$$

The rate law governing the adsorption process was also analyzed by using the moving boundary, film diffusion, and intraparticle diffusion mechanisms [26] (Equations (13)–(15)):

$$3 - 3 \times (1 - F)^{\frac{2}{3}} - 2 \times F = k \times t \tag{13}$$

$$\ln(1 - F) = -k \times t \tag{14}$$

$$\ln\left(1 - F^2\right) = -k \times t \tag{15}$$

where k refers to the corresponding constants and F is the factorial approach to equilibrium (Equation (16)):

$$F = \frac{[Tb]_t}{[Tb]_e} \tag{16}$$

where $[Tb]_t$ and $[Tb]_e$ are the concentrations of terbium adsorbed onto the activated carbon at time t and at equilibrium, respectively.

### 2.4. Desorption Experiments

Desorption experiments were also performed in a glass reactor provided with mechanical shaking at room temperature. To recover the terbium adsorbed onto the activated carbon, filtered Tb-loaded adsorbent (0.01 g Tb-AC) was added to 0.2 and 0.5 M HCl solutions. In addition, for a fixed Tb-AC dosage, different volumes of desorption solutions were investigated. After the corresponding elapsed time, the terbium concentration in the resulting solution as well as the desorption rate were assessed.

## 3. Results and Discussion

### 3.1. Characterization of the Obtained Adsorbent

The nitrogen adsorption–desorption isotherms at 77 K obtained for the AC are shown in Figure 2. A great increase in the quantity adsorbed at low pressures was found, which indicates the presence of microporosity in the material [27,28]. The quantity adsorbed at $p/p_0 = 1$ was 669 cm$^3$/g. Furthermore, a hysteresis loop can be observed. The lower part of the hysteresis loop represents the filling of the mesopores, and the upper part is attributed to the emptying [29]. According to the International Union of Pure and Applied Chemistry (IUPAC), the obtained shapes of the isotherms correspond to low $p/p_0$ for type I and high $p/p_0$ for type IV [30]. These shapes are typical of microporous and mesoporous materials, respectively. Thus, the obtained results revealed that the obtained material exhibits both the presence of micropores and mesopores. The calculated total pore volume was 1.03 cm$^3$/g, higher than the obtained 0.30 cm$^3$/g volume of micropores. Furthermore, the average micropore size was 4.2 nm, which corresponds to mesopores [31]. Again, these results revealed the presence of both micropores and mesopores in the carbon structure. Finally, the total specific area (Stot) values calculated using the Dubinin–Radushkevich equation [21] and the BET surface area (SBET) were 537 m$^2$/g and 981 m$^2$/g, respectively. It should be noted that the BET equation would seem inappropriate in the present case due to the high pore size, which leads to overestimation in the BET results [16]. However, the obtained calculated value is displayed for comparison purposes.

The elemental chemical compositions of the spent coffee grounds (SCG) and the final activated carbon (AC) are exhibited in Table 1. An increase in the carbon content as a consequence of the carbonization process can be appreciated in the AC sample.

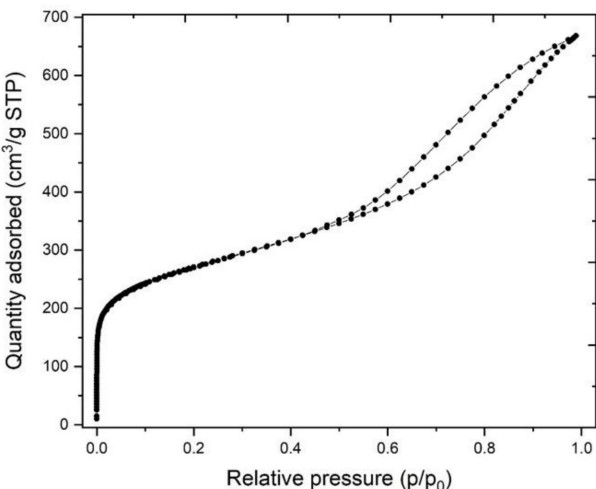

**Figure 2.** Nitrogen adsorption–desorption isotherms for the obtained AC.

**Table 1.** Composition of the precursor (SCG) and the final AC.

| Analysis | SCG | AC |
|:---:|:---:|:---:|
| C (wt% daf) | 50.7 | 85.4 |
| H (wt% daf) | 7.9 | 0.9 |
| N (wt% daf) | 2.5 | 0.7 |
| S (wt% daf) | 0.1 | 0.1 |
| O [a] (wt% daf) | 38.8 | 12.9 |

SCG: spent coffee grounds; daf: dry ash-free basis; [a] By subtraction.

The XRD patterns of the obtained activated carbon are shown in Figure 3. Two broad diffraction maxima at around 25° and 43° were found and can be attributed to the (0 0 2) and (1 0 0) planes of graphitic carbon. The peak at 43° indicates a certain degree of graphitization of the analyzed sample and the formation of a higher degree of intralayer condensation of graphite.

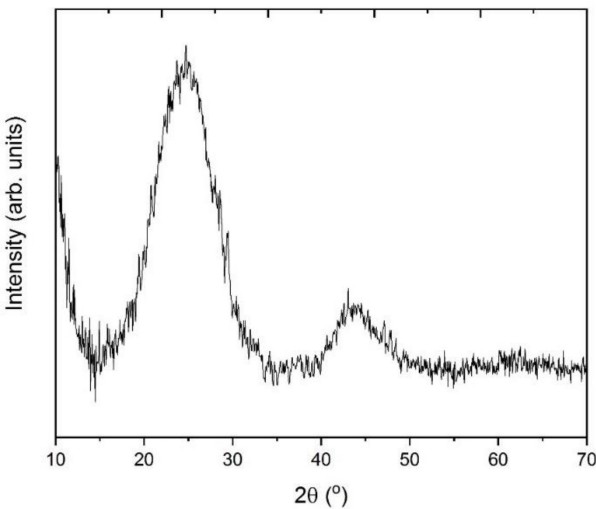

**Figure 3.** X-ray diffraction (XRD) pattern of the obtained AC.

Figure 4 shows the FE-SEM micrograph for the investigated sample. The obtained activated carbon shows heterogeneous macroporosity on the surface, with a low number of pores. In addition, a pitted and cracked surface was found. This surface is typical of physically activated carbons due to the treatment process [32]. However, at a higher

magnification (inset of Figure 4), a mesoporous microstructure can be appreciated. Pore sizes of less than 1 µm were found. This result is in good accordance with the results of BET measurements, which revealed the mesoporous structure of the material investigated.

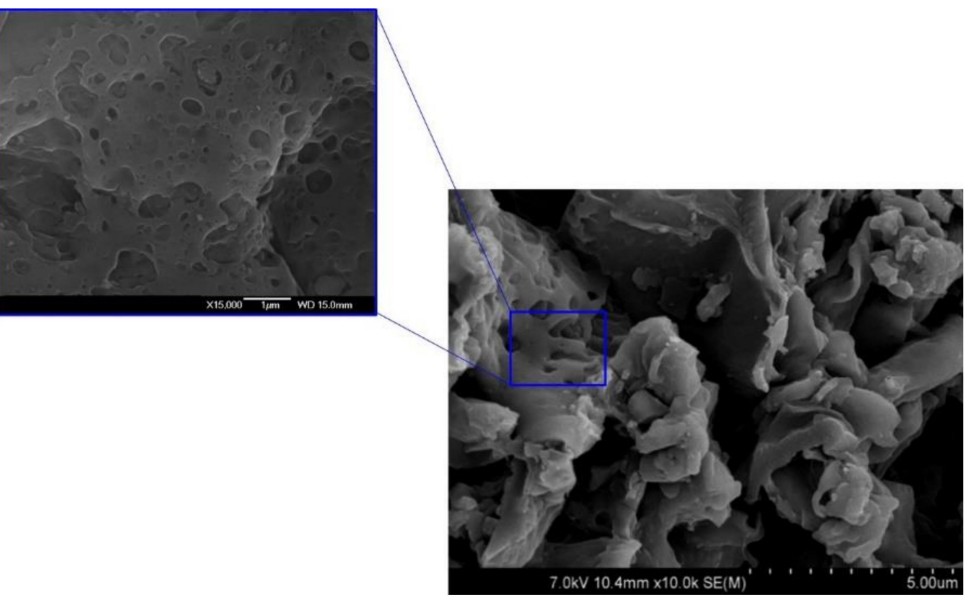

**Figure 4.** Field emission scanning electron microscopy (FE-SEM) image for the AC. A higher magnification micrograph is shown in the inset.

The micro-Raman spectrum is shown in Figure 5. The carbonaceous materials spectra significantly changed due to the type of abundant allotropic forms of carbon, as well as the fine structural changes of the individual allotrope [13]. Usually, graphitic materials spectra are characterized by two predominant bands, the so-called G-band and D-band, attributed to the $E_{2g}$ and $A_{1g}$ in-plane vibration modes, respectively [33]. The G-band is associated with the degree of disorder of the sample, while the D-band is related to the defects. Thus, the graphitization and disorder degree can be estimated by the intensity ratio of D and G band ($I_G/I_D$). In the present case, two sharp bands peaking at around 1600 cm$^{-1}$ and 1350 cm$^{-1}$ were found. Furthermore, the intensity of the D-band was slightly higher than that of the G-band, revealing a certain graphitization degree of the activated carbon.

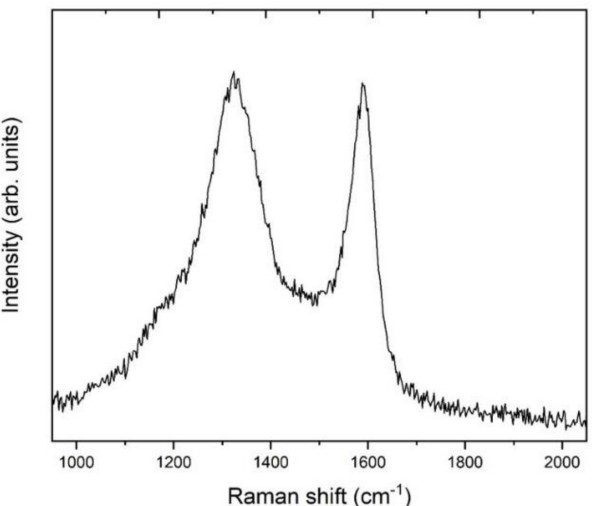

**Figure 5.** Registered micro-Raman spectrum for the obtained AC.

### 3.2. Terbium Adsorption Experiments

### 3.2.1. Influence of the Solution pH

To analyze the effect of pH on the terbium adsorption capacity with regard to the studied adsorbent, different experiments were carried out, modifying the solution pH value. This effect is important in adsorption processes because of the influences of the solution pH on the surface charge of the adsorbent and also on the species that the adsorbate will form in the solution. In the case of terbium, only $Tb^{3+}$ exists under acidic conditions and with near neutral pH values [34]. Consequently, in this investigation the pH value was varied from 5 towards more acidic values.

The adsorption capacity as well as the adsorption percentage are shown in Figure 6. As can be observed, both values increased with the increase in the pH value. In general, ACs exhibited a low point of zero charge pH ($pH_{PZC}$). Previous investigations revealed that ACs obtained from cane sugar bagasse, palm pit, sawdust [35], coconut shell [36], winemaking wastes [15], or even commercial activated carbon [37] showed $pH_{PZC}$ values lower than 5. In the present case, the calculated point of zero charge ($pH_{PZC}$) for the investigated AC was 4.6 (see Figure 7). At pH values lower than the corresponding $pH_{PZC}$, the AC surface presented a positive charge, while at a higher pH than the $pH_{PZC}$ value the surface was negatively charged. The obtained results could indicate that the surface of this AC becomes predominately positively charged at pH 2–3, so there was electrostatic repulsion between the surface and the REE ions. Thus, the adsorption scarcely occurred, and an adsorption capacity of around 4.5 mg/g (with a percentage of 22.5%) was found (pH 2). Nevertheless, as the pH increased, the surface of the AC was negatively charged and the terbium adsorption improved. Thus, the $q_t$ values for pH 2, 3, 4, and 5 were 4.8 mg/g, 6.5 mg/g, 13 mg/g, and 17.5 mg/g, respectively (i.e., adsorption percentages of 32.5%, 65%, and 87.5%). Thus, the subsequent adsorption experiments were carried out at pH 5, where under the present experimental conditions the maximum adsorption was obtained.

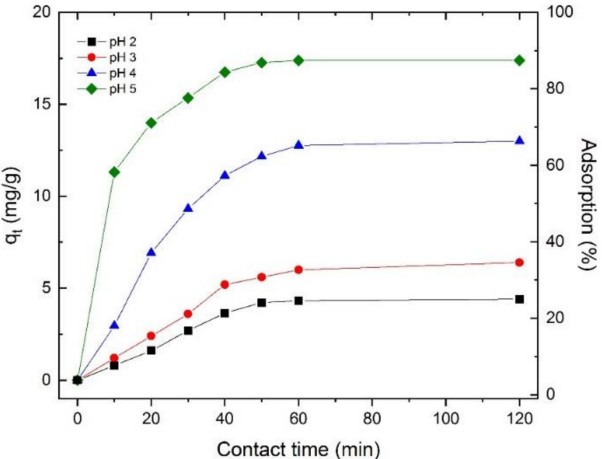

**Figure 6.** Influence of the solution pH on the terbium adsorption. Feed phase: 0.005 g/L Tb. Adsorbent dosage: 0.25 g/L. Temperature: 25 °C.

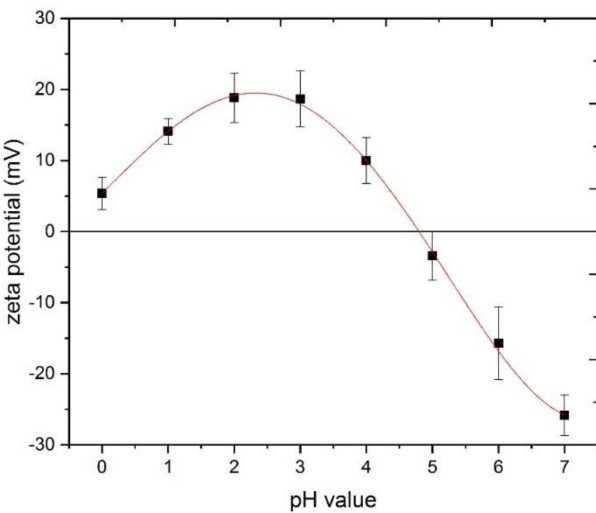

**Figure 7.** Zeta potential as a function of the pH values.

### 3.2.2. Influence of the Temperature

As the temperature at which the adsorption process is carried out could improve or hamper the process [38], several experiments at different temperatures were carried out. Adsorption capacities as a function of the contact time at the different temperatures analyzed are displayed in Figure 8. An increase in the adsorption capacity was found with the increase in temperature. This effect was more noticeable when the temperature increased from 25 °C to 40 °C, where the calculated $q_t$ value varied from 17.5 mg/g to 19.5 mg/g (i.e., an adsorption percentage increase from 87.5 to 97.7%). In addition, when the experimental temperature was increased to 60 °C, the terbium adsorption process was near quantitative, and an adsorption capacity of 20 mg/g was found. The obtained results revealed that the increase in temperature has practically no influence on the adsorption capacity. On the other hand, equilibrium was achieved earlier when the temperature increased. While the time of equilibrium to 25 °C was 60 min, the times of equilibrium to 40 °C and 60 °C were practically the same and equal to 40 min.

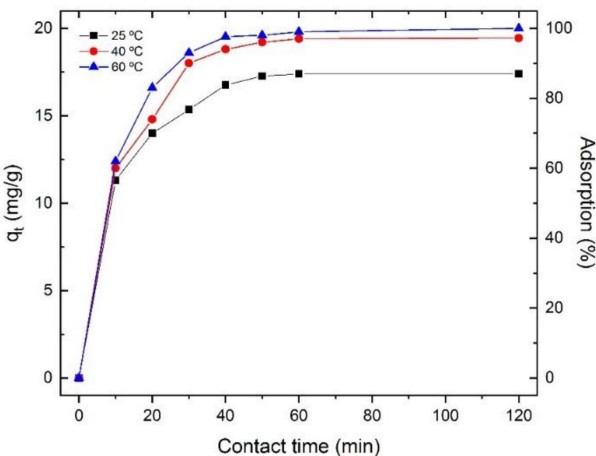

**Figure 8.** Influence of temperature on terbium adsorption. Other experimental conditions are as detailed in Figure 6.

### 3.2.3. Influence of the Adsorbent Dosage

Different adsorbent dosages were evaluated for the terbium adsorption process. As can be appreciated from Figure 9, terbium adsorption percentages increased with the adsorbent dosage for a fixed REE concentration in the same experimental conditions. In addition,

for adsorbent amounts greater than 0.5 g/L the terbium adsorption was considered as near quantitative.

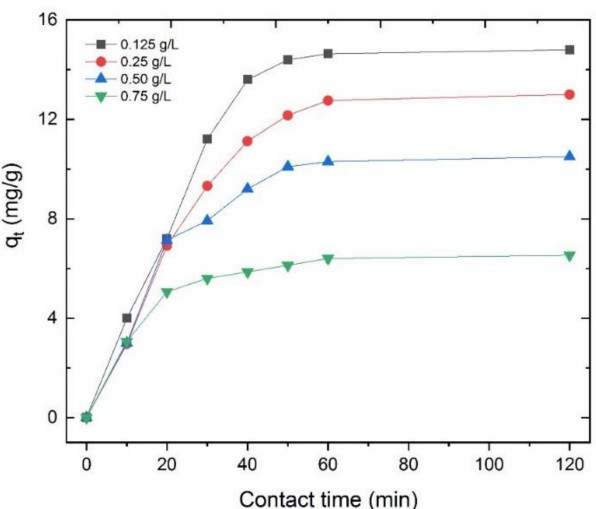

**Figure 9.** Influence of adsorbent concentration in the terbium adsorption process. Feed phase: 0.005 g/L Tb at pH 5. Temperature: 25 °C.

### 3.2.4. Fitting the Experimental Data to Adsorption Isotherms, Kinetics, and Rate Law Models

The equilibrium isotherms analyzed were: type-1 Langmuir, type-2 Langmuir, Freundlich, and Temkin (Equations (3)–(6)), all in their linear forms. Figure 10 exhibit the corresponding fittings of the experimental data.

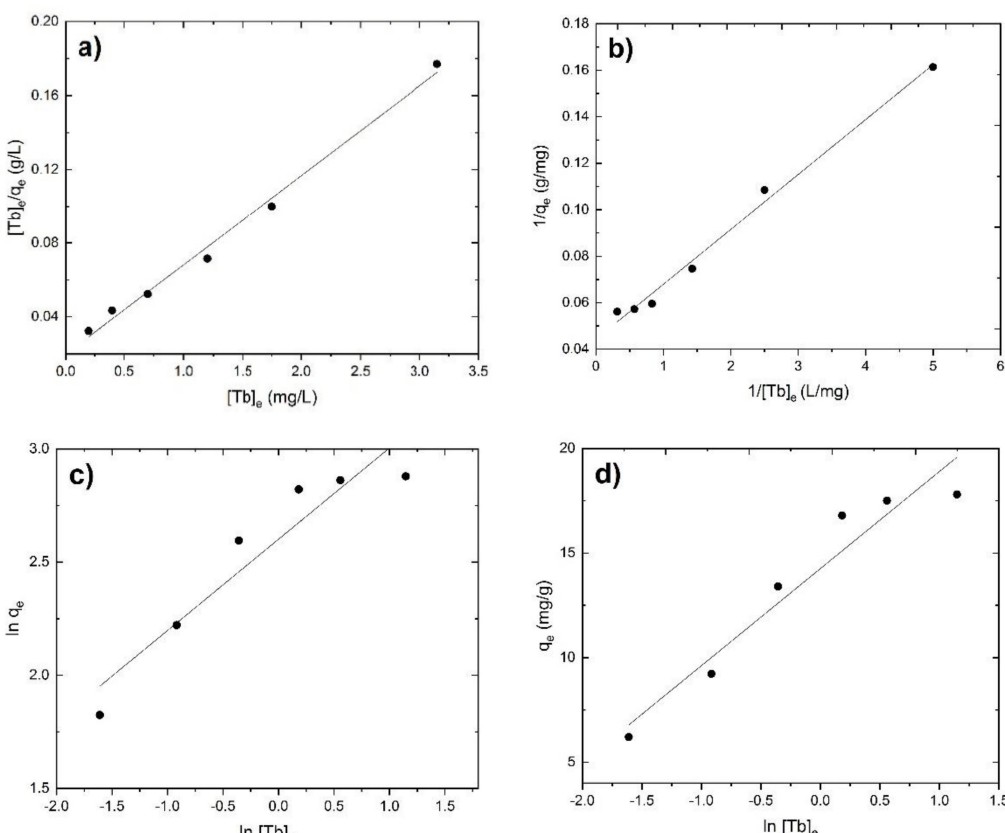

**Figure 10.** Fitting of (**a**) type-1 Langmuir, (**b**) type-2 Langmuir, (**c**) Freundlich, and (**d**) Temkin adsorption isotherm models.

The respective correlation coefficients of the fitting of the experimental Tb-loading data (Section 3.2.3) to these isotherm models (Table 2) showed that the type-1 Langmuir isotherm best described the terbium(III) adsorption process. Moreover, the equilibrium parameter $R_L$ (dimensionless) was calculated as:

$$R_L = \frac{1}{1 + b \times [Tb]_0} \tag{17}$$

where b is the Langmuir constant and $[Tb]_0$ is the initial terbium concentration in the aqueous solution. The above equation indicated whether the adsorption process was favorable ($0 < R_L < 1$), unfavorable ($R_L > 1$), linear ($R_L = 1$), or irreversible ($R_L = 0$). In the present case, the calculated $R_L$ constant was 0.03, indicating that the adsorption process was favorable.

**Table 2.** Calculated parameters and correlation coefficients for the equilibrium isotherms.

| Type-1 Langmuir | | Type-2 Langmuir | | Freundlich | | Temkin | |
|---|---|---|---|---|---|---|---|
| $q_m$ (mg/g) | 14.9 | $q_m$ (mg/g) | 44.5 | $k_F$ (L/g) | 12.0 | $A_T$ (L/g) | 1.2 |
| b (L/mg) | 8.0 | b (L/mg) | 0.2 | n (dimensionless) | 7.3 | $b_T$ (kJ/mol) | 200 |
| $R^2$ | 0.9932 | $R^2$ | 0.7679 | $R^2$ | 0.9722 | $R^2$ | 0.9286 |

The experimental equilibrium isotherm is displayed in Figure 11. A plateau can be observed, which confirms that the surface of the activated carbon is homogeneous. In addition, this result indicates that all sites on the surface have equal energy, and the terbium adsorption process onto the activated carbon could occur as one layer, as assumed by the Langmuir isotherm.

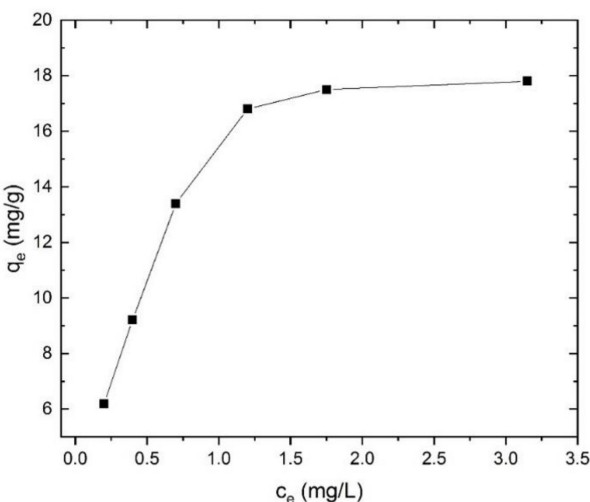

**Figure 11.** Experimental $Tb^{3+}$ loading isotherm. Temperature: 25 °C.

To analyze the kinetic order associated with the terbium adsorption process, the experimental data from Section 3.2.2 were plotted and fitted (Figure 12) to the corresponding models (Equations (7)–(9)). Table 3 shows the various calculated parameters as well as the derived correlation coefficients. For temperatures in the 25–60 °C range, the best fit of the experimental data was found to be to the pseudo-second-order kinetic model. Thus, an increase in the kinetic constant ($k_{2,calc}$) and the equilibrium adsorption capacity ($q_{e,calc}$) was found with the increase in temperature. Finally, it worth noting here that at the three temperatures the calculated adsorption capacities were similar to the experimental adsorption capacity $q_{e,exp}$ also shown in Table 3 for comparison purposes.

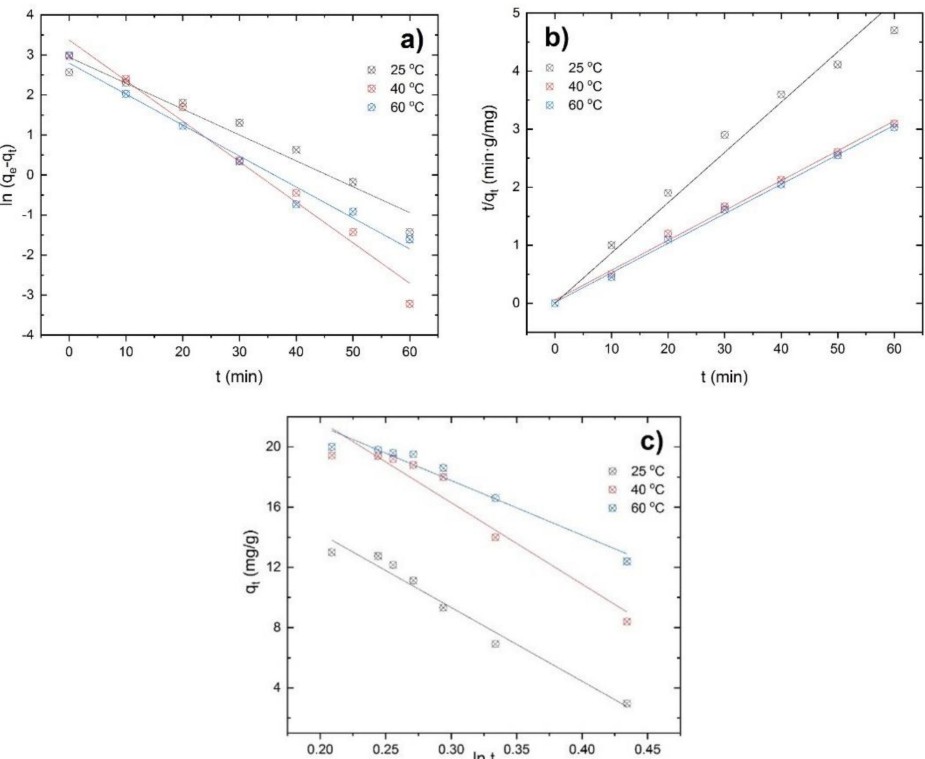

**Figure 12.** The fittings of adsorption kinetics: (**a**) linear pseudo-first-order model, (**b**) linear pseudo-second-order model, (**c**) Elovich model.

**Table 3.** Calculated kinetic parameters and correlation coefficients for the different temperatures analyzed.

| Temperature | $R^2$ | $k_{1,calc}$ (min$^{-1}$) | $q_{e,calc}$(mg/g) | $R^2$ | $k_{2,calc}$(g/mg·min) | $q_{e,calc}$ (mg/g) | $q_{e,exp}$ (mg/g) |
|---|---|---|---|---|---|---|---|
| 25 °C | 0.9523 | 0.066 | 18.9 | 0.9869 | 0.012 | 13.5 | 13.0 |
| 40 °C | 0.9770 | 0.101 | 29.1 | 0.9995 | 0.018 | 19.6 | 19.4 |
| 60 °C | 0.9807 | 0.077 | 16.4 | 0.9996 | 0.026 | 20.1 | 20.0 |

With the calculated kinetic constants, it is possible to learn the activation energy of the process using the Arrhenius equation (Equation (10)). Usually, the magnitude of the activation energy is used to establish if the adsorption is through a physisorption or chemisorption process. For the physisorption process, the energy required is smaller than that of the chemisorption process (5–40 kJ/mol versus 40–800 kJ/mol, respectively) [39]. In the present system, the calculated $E_a$ value was around 18 kJ/mol. This result suggests that the terbium adsorption onto this activated carbon is best represented by a physisorption process.

In addition, the thermodynamic parameters $\Delta H^0$, $\Delta S^0$, and $\Delta G^0$ were calculated using Equations (11) and (12). The positive value calculated for $\Delta H^0$ (114 kJ/mol) revealed the endothermic nature of the process. This result was in good agreement with the previous results that an increase of the temperature improved terbium uptake onto the AC. A positive value for the $\Delta S^0$ was found (0.4 kJ/mol·K), which indicated an increase in the randomness. Furthermore, $\Delta G^0$ was negative independently of the temperature (−6 kJ/mol, −12 kJ/mol, and −20 kJ/mol for 25 °C, 40 °C, and 60 °C, respectively), indicating that terbium adsorption is a spontaneous process.

Finally, the probable rate law governing the adsorption process was evaluated using Equations (13)–(15). As can be appreciated in Table 4, the best fit (with a correlation coefficient of 0.9919) was found within the moving boundary model and a calculated rate constant of 0.0165 min$^{-1}$. This model considers a sharp boundary which separates a reacted shell from an unreacted core. Thus, the adsorption occurred from the surface toward the center of the adsorbent with the progression of adsorption.

**Table 4.** Calculate rate constant and correlation coefficients for the rate law models.

| Rate Law Models | K (min$^{-1}$) | R$^2$ |
|---|---|---|
| Moving boundary | 0.0165 | 0.9919 |
| Film diffusion | 0.0632 | 0.9354 |
| Intraparticle diffusion | 0.0722 | 0.9660 |

### 3.3. Desorption Process

The terbium desorption step was investigated using a 13 mg/g Tb-loaded adsorbent. These solid were put in contact with HCl solutions with concentrations of 0.2 M and 0.5 M, and the eluent volume per mass of adsorbent was analyzed. The results (Table 5) show that when the eluent volume (mL)/mass of AC (g) relationships increased, the desorption percentage was practically the same for both HCl concentrations used in the desorption experiments. Nevertheless, when the eluent concentration increased from 0.2 M to 0.5 M, the rate of metal recovered in the eluate increased from 66% to 86%, respectively. These results reveal that it is possible for the terbium recovery efficiently from the loaded obtained activation carbon.

**Table 5.** Desorption results obtained from the different experiments.

| Conditions | Eluent/AC (mL/g) | Terbium Concentration in Solution (mg/L) | Desorption (%) |
|---|---|---|---|
| HCl (0.2 M) | 2500 | 3.48 | 67 |
|  | 5000 | 1.71 | 66 |
|  | 10,000 | 0.85 | 65 |
| HCl (0.5 M) | 2500 | 4.5 | 87 |
|  | 5000 | 2.2 | 85 |
|  | 10,000 | 1.1 | 85 |

## 4. Conclusions

In summary, activated carbon (AC) from spent coffee grounds (SCG) was obtained by a hydrothermal process followed by physical activation. The terbium adsorption onto the obtained AC as well as the effective recovery of the REE from aqueous solutions was proven. First, a hydro-alcoholic (50:50 ethanol:water (*v/v*)) suspension of the SCG was treated at 393 K for 30 min at 50 bars. Then, the obtained carbonaceous precursor was physically activated at 1073 K for 120 min, obtaining the final AC product.

The AC was deeply characterized. Nitrogen adsorption–desorption isotherms revealed the presence of both micropores and mesopores. In addition, a porous mesostructure was found in the corresponding scanning electron microscopy (SEM) images. Micro-Raman spectrum of the AC showed two principal bands characteristics of carbonaceous materials, with the D-band intensity slightly higher than the intensity of the G-band, revealing a certain graphitic character.

The as-obtained AC was used as an adsorbent for terbium, and several parameters that affect the process were analyzed. The adsorption capacity increased with the increase in the pH value (from 2 to 5), probably due to the surface charge of the obtained AC. In addition, an increase in the adsorption capacity was found with the temperature, indicating that is an endothermic process. The best fit of the experimental data corresponded to the type-1 Langmuir adsorption isotherm and to the pseudo-second-order kinetic model. In addition, thermodynamic studies revealed that the terbium adsorption process onto the obtained activated carbon was endothermic and spontaneous.

Finally, terbium was desorbed from the loaded–activated carbon under acidic conditions. Terbium desorption percentages of around 87% were found, indicating that effective terbium recovery was possible. Thus, from this final solution terbium may be recovered through carbonate or oxalate salts and further treatment to terbium metal.

**Author Contributions:** L.A., D.N.S., and F.J.A., methodology, validation, formal analysis, investigation, writing—original draft preparation, writing—review and editing; E.E., formal analysis; F.A.L., conceptualization, methodology investigation, writing—review and editing, funding acquisition, supervision, project administration. All authors have read and agreed to the published version of the manuscript.

**Funding:** This research has received funding from the European Union's Horizon 2020 research and innovation program under Grant No. 776851 (Car-E Service).

**Institutional Review Board Statement:** Not applicable.

**Informed Consent Statement:** Not applicable.

**Data Availability Statement:** Not applicable.

**Acknowledgments:** We acknowledge the support for the publication fee from the CSIC Open Access Publication Support Initiative through its Unit of Information Resources for Research (URICI). The authors would like thank to the Textural Characterization Service of the ICMM (CSIC) for the $N_2$ adsorption isotherms carried out.

**Conflicts of Interest:** The authors declare no conflict of interest.

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
