# Peer review of "Application of Activated Carbon Obtained from Spent Coffee Ground Wastes to Effective Terbium Recovery from Liquid Solutions"

_metals, doi:10.3390/met11040630_

Round 1
Reviewer 1 Report
I’ve just finished a review of the paper metals-1173502 titled: “Application of an activated carbon obtained from spent coffee ground wastes to the effective terbium recovery from liquid solutions” and written by authors: Lorena Alcaraz, Dayana Nathaly Saquinga, Francisco J. Alguacil, Esther Escudero and Félix A. López.
In the paper, authors investigated removal of terbium from liquid solutions by using activated carbon material obtained from spent coffee ground wastes, and investigated possibility of the terbium recovery after its removal from liquid solution. In the paper, activated carbon was characterized in detail, and after that influence of different parameters on terbium removal were investigated. Authors conclude that it is possible to remove terbium from liquid solution by using applied material, as well as, effective terbium recovery.
In general, paper is very easy for reading, the topic is interesting, and experiments were well designed. The paper is in accordance with the journal topic and i recommend it acceptance for publication. However, although the results are good, the explanations related with adsorption experiments and results were not good, and from that point I suggest acceptance but after major revision.
Below are my other comments:
- In experimental section, authors should give exact information about amount of AC used in adsorption experiments.
- In 3.2.1 In line 228 authors should insert a value of the pHpzc of the AC used in this study. Authors just mentioned that pHpzc<5. That means, for pH2-5 surface of the AC possess positive charge, but for pH>5 surface of the AC possess negative charge but for pH=5 surface is electroneutral. Since authors did not mentioned a value of the pHpzc, it is not clear how did they conclude that at pH=5 surface of the AC is negative. Also, in this paragraph is missing information about presence, potential competence and influence of the H3O+ ions on effectiveness of the terbium removal, what also may be a reason for better removal on higher pH. If authors measured final pHs their increasing or decreasing may be good indication for that. Also, these measuring may give a suggestion about reaction mechanism, for example ion exchange what is in agreement with increasing of entropy. From that reason, it is required from authors to add new explanations, insert pHpzc value of the applied AC and make clear paragraph in lines 226-241. Since the experiment was nicely done, I would ask the authors to make an effort and explain such good results just a little better.
- In 3.2.2. is missing conclusion about influence of T on effectiveness of the terbium removal, i.e. is missing answer on question: why were recorded such small changes in effectiveness (about 10 %) when temperature increased? Such explanation and results authors must connect with thermodynamic parameters. For this paragraph is the same comment as for previous - Since the experiment was nicely done, I would ask the authors to make an effort and explain such good results just a little better.
- In 3.2.3. In Figure 6, replace percentages with adsorbed amount (mg/g or mmol/g). That graph probably will not be the straight line, and authors should give explanation for such trend.
- In line 279 the formulae for determination equilibrium parameter RL is missing. Also, in 3.2.4. based on value of RL parameter authors concluded that adsorption is included in terbium removal from liquid solution and pointed out that Langmuir's model best describes a given system. On the other side, in pHmetric measuring, authors concluded that surface charge and ions interactions have influence of terbium removal. From kinetic measuring, authors found that physisorption is responsible for terbium removal. Also, thermodynamic measuring showed increasing of entropy. All of these if only Langmuir model is considered is not in agreement, although all conclusions are correct. The answer is in beside Langmuir, very hight R2 for Freundlich isotherm, which in combination with Langmuir model may cover all mentioned conclusions. From that reason, authors should rearrange and refill and expand given explanations.
- In line 320 authors mentioned that ΔH is positive, but with value “-114 kJ/mol”. Authors should check value or change comment.
- In line 323 explanation is required for ΔS increasing after removal terbium from solution. If the ions are removed from solution, it is logical that disorder of the system should be lower. Increasing of ΔS may be an additional proof that releasing of something from surface occur. I believe that Ion exchange between terbium from solution and monovalent cations from surface beside adsorption probably occurs, what is in agreement with pH metric and all other presented adsorption results.
- What about removal mechanism may be concluded from results listed in Table 3?
- The title of the paper and desorption experiment are not in agreement. Namely, in desorption experiments is missing part which describe how is possible to recover terbium from solution after its desorption. On this way, terbium is removed from solution by adsorbent, and then again returned in solution by desorption, but its extraction from solution is missing. Authors should add additional explanations in this regard.
Finally, I must I commend the authors for very well planned experiments and very good results, and kindly ask them to make an effort and explain such good results just a little better.
Best regards
Author Response
The authors would like to thank the reviewer for his/her useful comments.

Reviewer 2 Report
Manuscript ID: metals-1173502
Title: Application of an activated carbon obtained from spent coffee ground wastes to the effective terbium recovery from liquid solutions
Authors: Lorena Alcaraz et al.
Introduction. The introduction provides a comparison of extraction methods for terbium and writes that some methods are expensive and long-time. How were these values compared? Can the authors indicate the numerical values in question? What methods are used now, what is the cost of obtaining terbium and the duration of these processes. What is the cost of reagents (e.g. resins or extractants)?
Line 71-72. What type of coffee was used in this research? Describe this information in detail.
Line 72-74. Why need high-pressure treatment of SCG? Why uses these conditions?
Sections 2.1 Add a flow chart to better understand methods for AC preparation.
Line 106. How was terbium solution prepared? What reagent was used?
Line 121-122. Why don’t use Sips equation?
Line 134-135. Why don’t authors use Elovich equation?
Equation 12. Why authors write this form? Usually use another form: 1 – 2/3X – (1–X)2/3.
Section 2. What the XRD, chemical composition and particle size distribution of AC?
Table 1. Why authors don’t add figure with all equilibrium isotherms?
Table 2. It is necessary to add figure with kinetics calculations. In section 2.3 authors write three equations. Why in Table 2 only two type of data? Where is k3?
The authors should add discussion on further extraction of terbium after desorption. What real solutions can be used to extract terbium using AC?
Conclusions should be divided into 3-5 paragraphs.
Technical errors:
Line 33. Add chemical formulas for dysprosium (Dy), europium (Eu), neodymium (Ne), terbium (Tb), and yttrium (Y).
Author Response

(The authors gave the same response as above.)

Round 2
Reviewer 1 Report
Dear authors,
The reviewed version of the manuscript is much more improved in comparison with previous. Authors made an effort and added additional explanations. I think that the paper should be accepted and published in present form.
Regards
Reviewer 2 Report
The authors have greatly improved the article. There were added new figures of kinetics calculations. The methods of experiments is now more visual as there is a diagram of the experiments.
In this form, the article "Application of an activated carbon obtained from spent coffee ground wastes to the effective terbium recovery from liquid solutions" can be accepted into Metals.